# Neural Algorithmic Reasoners are Implicit Planners

**Andreea Deac**[*]
Mila – Québec AI Institute
Université de Montréal

**Petar Veličković**
DeepMind

**Ognjen Milinković**
Faculty of Mathematics
University of Belgrade

**Pierre-Luc Bacon**
Mila – Québec AI Institute
Université de Montréal

**Jian Tang**
Mila – Québec AI Institute
HEC Montréal

**Mladen Nikolić**
Faculty of Mathematics
University of Belgrade

## Abstract

Implicit planning has emerged as an elegant technique for combining learned models of the world with end-to-end model-free reinforcement learning. We study the class of implicit planners inspired by *value iteration*, an algorithm that is guaranteed to yield perfect policies in fully-specified tabular environments. We find that prior approaches either assume that the environment is provided in such a tabular form—which is highly restrictive—or infer "local neighbourhoods" of states to run value iteration over—for which we discover an algorithmic bottleneck effect. This effect is caused by explicitly running the planning algorithm based on scalar predictions in every state, which can be harmful to data efficiency if such scalars are improperly predicted. We propose eXecuted Latent Value Iteration Networks (XLVINs), which alleviate the above limitations. Our method performs all planning computations in a high-dimensional *latent space*, breaking the algorithmic bottleneck. It maintains alignment with value iteration by carefully leveraging neural graph-algorithmic reasoning and contrastive self-supervised learning. Across eight low-data settings—including classical control, navigation and Atari—XLVINs provide significant improvements to data efficiency against value iteration-based implicit planners, as well as relevant model-free baselines. Lastly, we empirically verify that XLVINs can closely align with value iteration.

## 1 Introduction

Planning is an important aspect of reinforcement learning (RL) algorithms, and planning algorithms are usually characterised by explicit modelling of the environment. Recently, several approaches explore *implicit planning* [40, 30, 33, 37, 29, 17, 16]. Such approaches propose inductive biases in the policy function to enable planning to emerge, while training the policy in a model-free manner. Accordingly, implicit planners combine the effectiveness of large-scale neural network training with the data efficiency promises of planning, making them a very attractive research direction.

Many popular implicit planners attempt to align with the computations of the value iteration (VI) algorithm within a policy network [40, 30, 29, 13, 26]. As VI is a differentiable algorithm, guaranteed to find the *optimal* policy, it can combine with gradient-based optimisation and provides useful theoretical guarantees. We also recognise the potential of VI-inspired deep RL, hence it is our primary topic of study here. However, applying VI assumes that the underlying RL environment (a) is *tabular*, and that its (b) *transition* and (c) *reward* distributions are both **fully known** and provided upfront. Such assumptions are unfortunately unrealistic for most environments of importance to RL research. Very often the dynamics of the environment will not be even partially known, and the state space may

---

[*]Work performed while the author was at DeepMind.

either be continuous (e.g. for control tasks) or very high-dimensional (e.g. for pixel-space observation in Atari), making a tabular representation hard to realise from a storage complexity perspective.

Accordingly, VI-based implicit planners often offer representation learning based solutions for alleviating some of the above limitations. Impactful early work [40, 29, 26] showed that, in tabular settings with known transition dynamics, the reward distribution and VI computations can be approximated by a (graph) convolutional network. While highly insightful, this line of work still does not allow for RL in generic non-tabular environments with unobserved dynamics. Conversely, approaches such as ATreeC [13] and VPN [30] lift the remaining two requirements, by using a latent transition model to construct a "local environment" around the current state. They then use learned models to predict scalar rewards and values in every node of this environment, applying VI-style algorithms directly.

While such approaches apparently allow for seamless VI-based implicit planning, we discover that the prediction of scalar signals represents an *algorithmic bottleneck*: if the neural network has observed insufficient data to properly estimate these scalars, the predictions of the VI algorithm will be equally suboptimal. This is limiting in low-data regimes, and can be seen as unfavourable, particularly given that one of the main premises of implicit planning is improved data efficiency.

In this paper, we propose the **eXecuted Latent Value Iteration Network** (XLVIN), an implicit planning policy network which embodies the computation of VI while addressing *all* of the limitations mentioned previously. We retain the favourable properties of prior methods while simultaneously performing VI in a high-dimensional latent space, removing the requirement of predicting scalars and hence *breaking* the algorithmic bottleneck. We enable this high-dimensional VI execution by leveraging the latest advances in neural algorithmic reasoning [43]. This emerging area of research seeks to emulate iterations of classical algorithms (such as VI) directly within neural networks. As a result, we are able to seamlessly run XLVINs with minimal configuration changes on a wide variety of discrete-action environments, including pixel-based ones (such as Atari), fully continuous-state control and navigation. Empirically, the XLVIN agent proves favourable in low-data environments against relevant model-free baselines as well as the ATreeC family of models.

Our contributions are thus three-fold: (a) we provide a detailed overview of the prior art in value iteration-based implicit planning, and discover an *algorithmic bottleneck* in impactful prior work; (b) we propose the XLVIN implicit planner, which breaks the algorithmic bottleneck while retaining the favourable properties of prior work; (c) we demonstrate a successful application of neural algorithmic reasoning within reinforcement learning, both in terms of quantitative analysis of XLVIN's data efficiency in low-data environments, and qualitative alignment to VI.

## 2 Background and related work

We will now present the context of our work, by gradually surveying the key developments which bring VI into the implicit planning domain and introducing the building blocks of our XLVIN agent.

**Planning** has been studied under the umbrella of model-based RL [36, 34, 20]. However, having a good model of the environment's dynamics is essential before being able to construct a good plan. We are instead interested in leveraging the progress of model-free RL [35, 27] by enabling planning through inductive biases in the policy network—a direction known as implicit planning. The planner could also be trained to optimise a supervised imitation learning objective [38, 2]. This is performed by UPNs [38] in a goal-conditioned setting. Our differentiable executors are instead applicable across a wide variety of domains where goals are not known upfront. Diff-MPC [2] leverages an algorithm in an explicit manner. However, explicit use of the algorithm often has issues of requiring a bespoke backpropagation rule, and the associated low-dimensional bottlenecks.

Throughout this section we pay special attention to implicit planners based on VI, and distinguish two categories of previously proposed planners: models which assume fixed and known environment dynamics [40, 29, 26] and models which derive scalars to be used for VI-style updates [13, 30].

**Value iteration (VI)** is a successive approximation method for finding the optimal value function of a discounted *Markov decision process* (MDPs) as the fixed-point of the so-called Bellman optimality operator [32]. A discounted MDP is a tuple $(\mathcal{S}, \mathcal{A}, R, P, \gamma)$ where $s \in \mathcal{S}$ are *states*, $a \in \mathcal{A}$ are *actions*, $R : \mathcal{S} \times \mathcal{A} \to \mathbb{R}$ is a reward function, $P : \mathcal{S} \times \mathcal{A} \to \text{Dist}(\mathcal{S})$ is a *transition function* such that $P(s'|s, a)$ is the conditional probability of transitioning to state $s'$ when the agent executes action $a$ in state $s$, and $\gamma \in [0, 1]$ is a discount factor which trades off between the relevance of immediate and

future rewards. In the infinite horizon discounted setting, an agent sequentially chooses actions according to a stationary Markov *policy* $\pi : \mathcal{S} \times \mathcal{A} \to [0, 1]$ such that $\pi(a|s)$ is a conditional probability distribution over actions given a state. The *return* is defined as $G_t = \sum_{k=0}^{\infty} \gamma^k R(a_{t+k}, s_{t+k})$. Value functions $V^\pi(s, a) = \mathbb{E}_\pi[G_t|s_t = s]$ and $Q^\pi(s, a) = \mathbb{E}_\pi[G_t|s_t = s, a_t = a]$ represent the expected return induced by a policy in an MDP when conditioned on a state or state-action pair respectively. In the infinite horizon discounted setting, we know that there exists an optimal stationary Markov policy $\pi^*$ such that for any policy $\pi$ it holds that $V^{\pi^*}(s) \geq V^\pi(s)$ for all $s \in \mathcal{S}$. Furthermore, such optimal policy can be deterministic – *greedy* – with respect to the optimal values. Therefore, to find a $\pi^*$ it suffices to find the unique optimal value function $V^\star$ as the fixed-point of the Bellman optimality operator. The optimal value function $V^\star$ is such a fixed-point and satisfies the *Bellman optimality equations* [7]: $V^\star(s) = \max_{a \in \mathcal{A}} \left( R(s, a) + \gamma \sum_{s' \in \mathcal{S}} P(s'|s, a)V^\star(s') \right)$. Accordingly, VI randomly initialises a value function $V_0(s)$, and then iteratively updates it as follows:

$$V_{t+1}(s) = \max_{a \in \mathcal{A}} \left( R(s, a) + \gamma \sum_{s' \in \mathcal{S}} P(s'|s, a)V_t(s') \right) . \tag{1}$$

VI is thus a powerful technique for optimal control in RL tasks, but its applicability hinges on knowing the MDP parameters (especially $P$ and $R$) upfront—which is unfortunately not the case in most environments of interest. To make VI more broadly applicable, we need to leverage function approximators (such as neural networks) and representation learning to estimate such parameters.

**Value iteration is message passing** Progress towards broader applicability started by lifting the requirement of knowing $R$. Several implicit planners, including (G)VIN [40, 29] and GPPN [26], were proposed for discrete environments where $P$ is fixed and known. Observing the VI update rule (Equation 1), we may conclude that it derives values by considering features of *neighbouring* states; i.e. the value $V(s)$ is updated based on states $s'$ for which $P(s'|s, a) > 0$ for some $a$. Accordingly, it tightly aligns with *message passing* over the graph corresponding to the MDP, and hence a graph neural network (GNN) [15] over the MDP graph may be used to estimate the value function.

**Graph neural networks** (GNNs) have been intensively studied as a tool to process graph-structured inputs, and were successfully applied to various RL tasks [45, 24, 1]. For each state $s$ in the graph, a set of messages is computed—one message for each neighbouring node $s'$, derived by applying a *message function* $M$ to the relevant node $(\vec{h}_s, \vec{h}_{s'})$ and edge $(\vec{e}_{s' \to s})$ features. Incoming messages in a neighbourhood $\mathcal{N}(s)$ are then aggregated through a permutation-invariant operator $\bigoplus$ (such as sum or max), obtaining a summarised message $\vec{m}_s$:

$$\vec{m}_s = \bigoplus\nolimits_{s' \in \mathcal{N}(s)} M(\vec{h}_s, \vec{h}_{s'}, \vec{e}_{s' \to s}) \tag{2}$$

The features of state $s$ are then updated through a function $U$ applied to these summarised messages:

$$\vec{h}'_s = U(\vec{h}_s, \vec{m}_s) \tag{3}$$

GNNs can then emulate VI computations by setting the neighbourhoods according to the transitions in $P$; that is, $\mathcal{N}(s) = \{s' \mid \exists a \in \mathcal{A}. \ P(s'|s, a) > 0\}$. For the special case of grid worlds, the neighbours of a grid cell correspond to exactly its neighbouring cells, and hence the rules in Equations 2–3 amount to a convolutional neural network over the grid [40].

While the above blueprint yielded many popular implicit planners, the requirement of knowing upfront the transition neighbourhoods is still limiting. Ideally, we would like to be able to, on-the-fly, generate states $s'$ that are reachable from $s$ as a result of applying some action. Generating neighbouring state representations corresponds to a learned *transition model*, and we present one such method, which we employed within XLVIN, next.

**TransE** The TransE [8] loss for embedding objects and relations can be adapted to RL [23, 42]. State embeddings are obtained by an *encoder* $z : \mathcal{S} \to \mathbb{R}^k$ and the effect of an action in a given state is modelled by a translation model $T : \mathbb{R}^k \times \mathcal{A} \to \mathbb{R}^k$. Specifically, $T(z(s), a)$ is a *translation vector* to be added to $z(s)$ in order to obtain an embedding of the resulting state when taking action $a$ in state $s$. This embedding should be as close as possible to $z(s')$, for the observed transition $(s, a, s')$, and also far away from negatively sampled state embeddings $z(\tilde{s})$. Therefore, the embedding function is optimised using the following variant of the triplet loss (with hinge hyperparameter $\xi$):

$$\mathcal{L}_{\text{TransE}}((s, a, s'), \tilde{s}) = d(z(s) + T(z(s), a), z(s')) + \max(0, \xi - d(z(\tilde{s}), z(s'))) \tag{4}$$

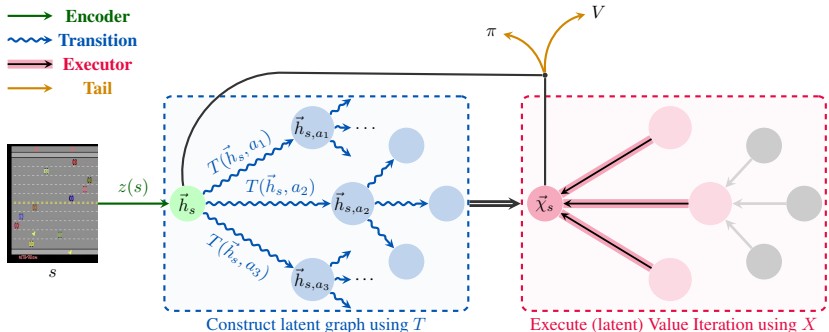

Figure 1: XLVIN model summary. Its modules are explained (and colour-coded) in Section 3.1.

Having a trained $T$ function, it is now possible to dynamically construct $\mathcal{N}(s)$. For every action $a$, applying $T(\vec{h}_s, a)$ yields embeddings $\vec{h}_{s,a}$ which correspond to one neighbour state embedding in $\mathcal{N}(s)$. We can, of course, roll out $T$ further from $\vec{h}_{s,a}$ to simulate longer trajectory outcomes—the amount of steps we do this for is denoted as the "thinking time", $K$, of the planner. With neighbourhoods constructed, one final question remains: how to apply VI over them, especially given that the embeddings $\vec{h}_s$ are high-dimensional, and VI is defined over scalar reward/value inputs?

If we also train a reward model, $R(\vec{h}_s, a)$, and a state-value function, $V(\vec{h}_s)$ from state embeddings, we can attach scalar values and rewards to our synthesised graph. Then, VI can be directly applied over the constructed tree to yield the final policy. As the VI update (Equation 1) is differentiable, it composes nicely with neural estimators and standard RL loss functions. Further, $R$ and $V$ can be directly trained from observed rewards and returns when interacting with the environment. This approach inspired a family of powerful implicit planners, including VPN [30], TreeQN and ATreeC [13]. However, it remains vulnerable to a specific bottleneck effect, which we discuss next.

**Algorithmic bottleneck** Through our efforts of learning transition and reward models, we reduced our (potentially highly complex) input state $s$ into an abstractified graph with scalar values in its nodes and edges, so that VI can be directly applied. However, VI's performance guarantees rely on having the *exactly correct* parameters of the underlying MDP. If there are any errors in the predictions of these scalars, they may propagate to the VI operations and yield suboptimal policies. We will study this effect in detail on synthetic environments in Section 4.3.

As the ATreeC-style approaches commit to using the scalar produced by their reward models, there is no way to recover from a poorly predicted value. This leaves the model vulnerable to an *algorithmic bottleneck*, especially early on during training when insufficient experience has been gathered to properly estimate $R$. Accordingly, the agent may struggle with data efficiency.

With our XLVIN agent, we set out to break this bottleneck, and do not project our state embeddings $\vec{h}_s$ further to a low-dimensional space. This amounts to running a graph neural network directly over them. Given that these same embeddings are optimised to produce plausible graphs (via the TransE loss), how can we ensure that our GNN will stay aligned with VI computations?

**Algorithmic reasoning** An important research direction explores the use of neural networks for learning to execute algorithms [10, 43]—which was recently extended to algorithms on graph-structured data [44]. In particular, [47] establishes *algorithmic alignment* between GNNs and dynamic programming algorithms. Furthermore, [44] show that supervising the GNN on the algorithm's intermediate results is highly beneficial for out-of-distribution generalization. As VI is, in fact, a dynamic programming algorithm, a GNN executor is a suitable choice for learning it, and good results on executing VI emerged on synthetic graphs [11]—an observation we strongly leverage here.

## 3 XLVIN Architecture

Next, we specify the computations of the eXecuted Latent Value Iteration Network (XLVIN). We recommend referring to Figure 1 for a visualisation of the model dataflow (more compact overview in Appendix A) and to Algorithm 1 for a step-by-step description of the forward pass.

We propose a *policy network*—a function, $\pi_\theta(a|s)$, which for a given state $s \in \mathcal{S}$ specifies a probability distribution of performing each action $a \in \mathcal{A}$ in that state. Here, $\theta$ are the policy parameters, to be optimised with gradient ascent.

## 3.1 XLVIN modules

**Encoder**  The encoder function, $z : \mathcal{S} \to \mathbb{R}^k$, consumes state representations $s \in \mathcal{S}$ and produces flat embeddings, $\vec{h}_s = z(s) \in \mathbb{R}^k$. The design of this component is flexible and may be dependent on the structure present in states. For example, pixel-based environments will necessitate CNN encoders, while environments with flat observations are likely to be amenable to MLP encoders.

**Transition**  The transition function, $T : \mathbb{R}^k \times \mathcal{A} \to \mathbb{R}^k$, models the effects of taking actions, in the *latent* space. Accordingly, it consumes a state embedding $z(s)$ and an action $a$ and produces the appropriate *translation* of the state embedding, to match the embedding of the successor state (in expectation). That is, it is desirable that $T$ satisfies Equation 5 and it is commonly realised as an MLP.

$$z(s) + T(z(s), a) \approx \mathbb{E}_{s' \sim P(s'|s,a)} z(s') \tag{5}$$

**Executor**  The executor function, $X : \mathbb{R}^k \times \mathbb{R}^{|\mathcal{A}| \times k} \to \mathbb{R}^k$, processes an embedding $\vec{h}_s$ of a given state $s$, alongside a neighbourhood set $\mathcal{N}(\vec{h}_s)$, which contains (expected) embeddings of states that immediately neighbour $s$—for example, through taking actions. Hence,

$$\mathcal{N}(\vec{h}_s) \approx \left\{ \mathbb{E}_{s' \sim P(s'|s,a)} z(s') \right\}_{a \in \mathcal{A}} \tag{6}$$

The executor combines the neighbourhood set features to produce an updated embedding of state $s$, $\vec{\chi}_s = X(\vec{h}_s, \mathcal{N}(\vec{h}_s))$, which is mindful of the properties and structure of the neighbourhood. Ideally, $X$ would perform operations in the latent space which mimic the one-step behaviour of VI, allowing for the model to meaningfully plan from state $s$ by stacking several layers of $X$ (with $K$ layers allowing for exploring length-$K$ trajectories). Given the relational structure of a state and its neighbours, the executor is commonly realised as a graph neural network (GNN).

**Actor & Tail components**  The actor function, $A : \mathbb{R}^k \times \mathbb{R}^k \to [0, 1]^{|\mathcal{A}|}$ consumes the state embedding $\vec{h}_s$ and the updated state embedding $\vec{\chi}_s$, producing action probabilities $\pi_\theta(a|s) = A\left(\vec{h}_s, \vec{\chi}_s\right)_a$, specifying the policy to be followed by our XLVIN agent. Lastly, note that we may also have additional *tail* networks which have the same input as $A$. For example, we train XLVINs using proximal policy optimisation (PPO) [35], which necessitates a state-value function: $V(\vec{h}_s, \vec{\chi}_s)$.

---

**Algorithm 1:** XLVIN forward pass

**Input**  :Input state $s$, executor depth $K$
**Output** :Policy function $\pi_\theta(a|s)$, state-value function $V(s)$

$\vec{h}_s = z(s)$ ;                                                  // Embed the input state with the encoder
$\mathbb{S}^0 = \{\vec{h}_s\}, \mathbb{E} = \emptyset$
**for** $k \in [0, K)$ **do**
$\quad$ $\mathbb{S}^{k+1} = \emptyset$ ;                                // Initialise depth-$(k+1)$ embeddings
$\quad$ **for** $\vec{h} \in \mathbb{S}^k$, $a \in \mathcal{A}$ **do**
$\quad\quad$ $\vec{h}' = \vec{h} + T(\vec{h}, a)$ ;                     // Get (expected) neighbour embedding
$\quad\quad$ $\mathbb{S}^{k+1} = \mathbb{S}^{k+1} \cup \{\vec{h}'\}, \mathbb{E} = \mathbb{E} \cup \{(\vec{h}, \vec{h}', a)\}$ ;                     // Attach $\vec{h}'$ to the graph
$\quad$ **end**
**end**
/* Run the execution model over the graph specified by the nodes $\mathbb{S} = \bigcup_{k=0}^{K} \mathbb{S}^k$ and edges $\mathbb{E}$, by repeatedly
$\quad$ applying $\vec{h} = X(\vec{h}, \mathcal{N}(\vec{h}))$, for every embedding $\vec{h} \in \mathbb{S}$, for $K$ steps.                                 */
$\vec{\chi}_s = \text{EXECUTE}(\vec{h}_s, \bigcup_{k=0}^{K} \mathbb{S}^k, \mathbb{E}, X, K)$ ;     // See Appendix B for details on the EXECUTE function
/* Use the actor and tail to predict the policy and value functions from the (updated) state embedding of $s$ */
$\pi_\theta(s, \cdot) = A(\vec{h}_s, \vec{\chi}_s), V(s) = V(\vec{h}_s, \vec{\chi}_s)$

---

**Discussion** The entire procedure is end-to-end differentiable, does not impose any assumptions on the structure of the underlying MDP, and has the capacity to perform computations directly aligned with value iteration, while avoiding the algorithmic bottleneck. This achieves all of our initial aims.

The transition function produces state embeddings that correspond to the expectation of the successor state embedding over all possible outcomes (Equation 5). While taking expectations is an operation that aligns well with VI computations, it can pose limitations in the case of non-deterministic MDPs. This is because the obtained expected latent states may not be trivially further expandable, should we wish to plan further from them. One possible remedy we suggest is employing a probabilistic (e.g. variational) transition model from which we could repeatedly sample concrete next-state latents.

Our tree expansion strategy is *breadth-first*, which expands every action from every node, yielding $O(|\mathcal{A}|^K)$ time and space complexity. While this is prohibitive for scaling up $K$, we empirically found that performance plateaus by $K \leq 4$ for all studied environments, mirroring prior findings [13]. Even if these are shallow trees of states, we anticipate a compounding effect from optimising TransE together with PPO's value/policy heads. We defer allowing for deeper expansions and large action spaces to future work, but note that it will likely require a *rollout policy*, selecting actions to expand from a given state. For example, I2A [33] obtains a rollout policy by distilling the agent's policy network. Extensions to continuous actions could also be achieved by rollout policies, or discretising the action space by binning [41].

## 3.2 XLVIN Training

As discussed, the success of XLVIN relies on our transition function, $T$, constructing plausible graphs, and our executor function, $X$, reliably imitating VI steps in a high dimensional latent space. Accordingly, we train both of them using established methods: TransE for $T$ and [11] for $X$.

To optimise the neural network parameters $\theta$, we use proximal policy optimisation (PPO)[2] [35]. Note that the PPO gradients also flow into $T$ and $X$, which could *displace* them from the properties required by the above, leading to either poorly constructed graphs or lack of VI-aligned computation.

Without knowledge of the underlying MDP, we have no easy way of training the executor, $X$, online. We instead opt to *pre-train* the parameters of $X$ and *freeze* them, treating them as constants rather than parameters to be optimised. In brief, the executor pre-training proceeds by first **generating** a dataset of synthetic MDPs, according to some underlying graph distribution. Then, we **execute** the VI algorithm on these MDPs by iterating Equation 1, keeping track of intermediate values $V_t(s)$ at each step $t$, until convergence. Finally, we **supervise** a GNN (operating over the MDP transitions as edges) to receive $V_t(s)$—and all other parameters of the MDP—as inputs, and predict $V_{t+1}(s)$ (optimised using mean-squared error). Such a graph neural network has three parts: an *encoder*, mapping $V_t(s)$ to a latent representation, a *processor*, which performs a step of VI in the latent space, and a *decoder*, which decodes back $V_{t+1}(s)$ from the latent space. We only **retain** the *processor* as our executor function $X$, in order to avoid the algorithmic bottleneck in our architecture.

For the transition model, we found it sufficient to optimise TransE (Equation 4) using only on-policy trajectories. However, we do anticipate that some environments will require a careful tradeoff between exploration and exploitation for the data collection strategy for training the transition model.

Thus, after pre-training the GNN to predict one-step value iteration updates and freezing the processor, a step of the training algorithm corresponds to:

1. Sample on-policy rollouts (with multiple parallel actors acting for a fixed number of steps).
2. Based on the transitions in these rollouts, evaluate the PPO [35] and TransE (Equation 4) losses. Negative sample states for TransE, $\tilde{s}$, are randomly sampled from the rollouts.
3. Update the policy network's parameters, $\theta$, using the combined loss. It is defined, for a single rollout, $\mathcal{T} = \{(s_t, a_t, r_t, s_{t+1})\}_t$, as follows:

$$\mathcal{L}(\mathcal{T}) = \mathcal{L}_{\text{PPO}}(\mathcal{T}) + \lambda \sum_{i=1}^{|\mathcal{T}|} \mathcal{L}_{\text{TransE}}((s_i, a_i, s_{i+1}), \tilde{s}_i) \qquad \tilde{s}_i \sim \mathbb{P}(s|\mathcal{T}) \tag{7}$$

where we set $\lambda = 0.001$, and $\mathbb{P}(s|\mathcal{T})$ is the empirical distribution over the states in $\mathcal{T}$.

---

[2]We use the PPO implementation and hyperparameters from Kostrikov [25].

It should be highlighted that our approach can easily be modified to support value-based methods such as DQN [27]—merely by modifying the tail component of the network and the RL loss function.

## 4 Experiments

We now deploy XLVINs in *generic* discrete-action environments with unknown MDP dynamics (further details in Appendix C), and verify their potential as an implicit planner. Namely, we investigate whether XLVINs provide gains in data efficiency, by comparing them in low-data regimes against a relevant model-free PPO baseline, and ablating against the ATreeC implicit planner [13], which executes an explicit TD($\lambda$) backup instead of a latent-space executor, and is hence prone to algorithmic bottlenecks. All chosen environments were previously explicitly studied in the context of planning within deep reinforcement learning, and were identified as environments that benefit from planning computations in order to generalise [21, 13, 30, 22].

### 4.1 Experimental setup

**Common elements** On all environments, the transition function, $T$, is a three-layer MLP with layer normalisation [4] after the second layer. The executor, $X$, is, for all environments, identical to the message passing executor of [11]. We train the executor from completely random deterministic graphs—making no assumptions on the underlying environment's topology.

**Continuous-space** We focus on four OpenAI Gym environments [9]: classical continuous-state control tasks—CartPole, Acrobot and MountainCar, and a continuous-state spaceship navigation task, LunarLander. In all cases, we study data efficiency by presenting extremely limited data scenarios.

The encoder function is a three-layer MLP with ReLU activations, computing 50 output features and $F$ hidden features, where $F = 64$ for CartPole, $F = 32$ for Acrobot, $F = 16$ for MountainCar and $F = 64$ for LunarLander. The same hidden dimension is also used in the transition function MLP.

As before, we train our executor from random deterministic graphs. In this setting only, we also attempt to illustrate the potential benefits when the graph distribution is biased by our beliefs of the environment's topology. Namely, we attempt training the executor from synthetic graphs that imitate the dynamics of CartPole very crudely—the MDP graphs being binary trees where only certain leaves carry zero reward and are terminal. More details on the graph construction, for both of these approaches, is given in Appendix E. The same trained executor is then deployed across all environments, to demonstrate robustness to synthetic graph construction. For LunarLander, the XLVIN uses $K = 3$ executor layers; in all other cases, $K = 2$.

It is worthy to note that CartPole offers dense and frequent rewards—making it easy for policy gradient methods. We make the task challenging by sampling *only 10 trajectories* at the beginning, and not allowing any further interaction—beyond 100 epochs of training on this dataset. Conversely, the remaining environments are all sparse-reward, and known to be challenging for policy gradient methods. For these environments, we sample 5 trajectories at a time, twenty times during training (for a total of 100 trajectories) for Acrobot and MountainCar, and fifty times during training (for a total of 250 trajectories) for LunarLander.

**Pixel-space** Lastly, we investigate how XLVINs perform on high-dimensional pixel-based observations, using the Atari-2600 [6]. We focus on four games: Freeway, Alien, Enduro and H.E.R.O.. These environments encompass various aspects of complexity: sparse rewards (in Freeway), larger action spaces (18 actions for Alien and H.E.R.O.) and visually rich observations (changing time-of-day on Enduro) and long-range credit assignment (on H.E.R.O.). Further, we successfully *re-use* the executor trained on random deterministic graphs, showing its transfer capability across vastly different settings. We evaluate the agents' low-data performance by allowing only 1,000,000 observed transitions. We re-use exactly the environment and encoder from Kostrikov [25], and run the executor for $K = 2$ layers for Freeway and Enduro and $K = 1$ for Alien and H.E.R.O..

### 4.2 Results

In our results, we use **"XLVIN-CP"** to denote XLVIN executors pre-trained using CartPole-style synthetic graphs (where applicable), and **"XLVIN-R"** for pre-training them on random deterministic graphs. **"PPO"** denotes our baseline model-free agent; it has no transition/executor model, but

Table 1: Mean scores for low-data CartPole-v0, Acrobot-v1, MountainCar-v0 and LunarLander-v2, averaged over 100 episodes and five seeds.

| Agent | CartPole-v0 10 trajectories | Acrobot-v1 100 trajectories | MountainCar-v0 100 trajectories | LunarLander-v2 250 trajectories |
|---|---|---|---|---|
| PPO | $104.6$ $\pm$ 48.5 | $-500.0$ $\pm$ 0.0 | $-200.0$ $\pm$ 0.0 | $90.52$ $\pm$ 9.54 |
| ATreeC | $117.1$ $\pm$ 56.2 | $-500.0$ $\pm$ 0.0 | $-200.0$ $\pm$ 0.0 | $84.04$ $\pm$ 5.35 |
| XLVIN-R | $\mathbf{199.2}$ $\pm$ 1.6 | $-353.1$ $\pm$ 120.3 | $-185.6$ $\pm$ 8.1 | $\mathbf{99.34}$ $\pm$ 6.77 |
| XLVIN-CP | $\mathbf{195.2}$ $\pm$ 5.0 | $\mathbf{-245.4}$ $\pm$ 48.4 | $\mathbf{-168.9}$ $\pm$ 24.7 | N/A |

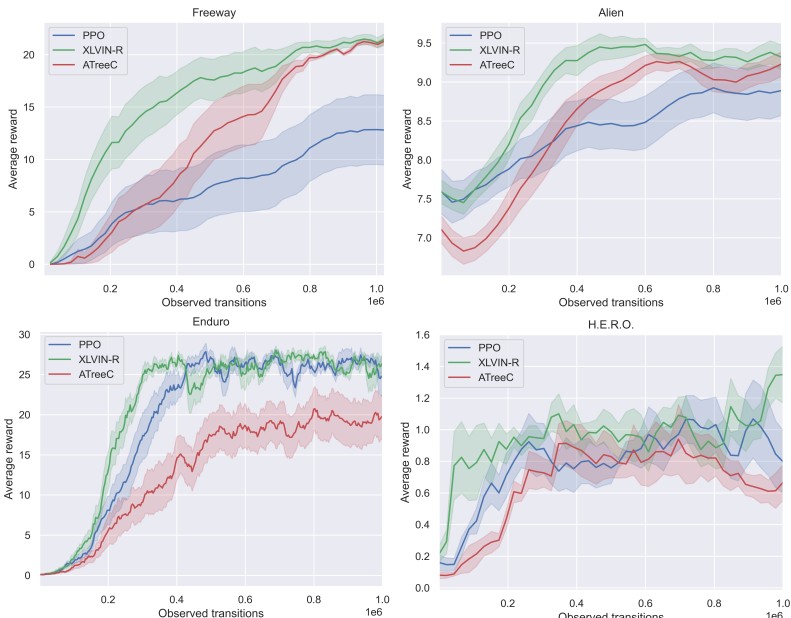

Figure 2: Average clipped reward on Freeway, Alien, Enduro and H.E.R.O. over 1,000,000 transitions and ten seeds.

otherwise matches the XLVIN hyperparameters. As planning-like computation was shown to emerge in entirely model-free agents [16], this serves as a check for the importance of the VI computation.

To analyse the impact of the algorithmic bottleneck, we use **"ATreeC"** [13] as one of our baselines, capturing the behavior of a larger class of VI-based implicit planners (including TreeQN and VPN [30]). For maximal comparability, we make ATreeC fully match XLVIN's hyperparameters, except for the executor model. Since ATreeC's policy is directly tied to the result of applying TD($\lambda$), its ultimate performance is closely tied to the quality of its scalar value predictions. Comparing against ATreeC can thus give insight into negative effects of the algorithmic bottleneck at low-data regimes.

**CartPole, Acrobot, MountainCar and LunarLander** Results for the continuous-space control environments are provided in Table 1. We find that the XLVIN model solves CartPole from only 10 trajectories, outperforming all the results given in [42] (incl. REINFORCE [46], Autoencoders, World Models [18], DeepMDP [14] and PRAE [42]), while using $10\times$ fewer samples. For more details, see Appendix D.

Further, our model is capable of solving the Acrobot and MountainCar environments from only 100 trajectories, in spite of sparse rewards. Conversely, the baseline model, as well as ATreeC, are unable to get off the ground at all, remaining stuck at the lowest possible score in the environment until timing out. This still holds when XLVIN is trained on the random deterministic graphs, demonstrating that the executor training need not be dependent on knowing the underlying MDP specifics.

Mirroring the above findings, XLVIN also demonstrates clear low-data efficiency gains on Lu-narLander, compared to the PPO baseline and ATreeC. In this setting, ATreeC even underperformed compared to the PPO baseline, highlighting once again the issues with algorithmic bottlenecks.

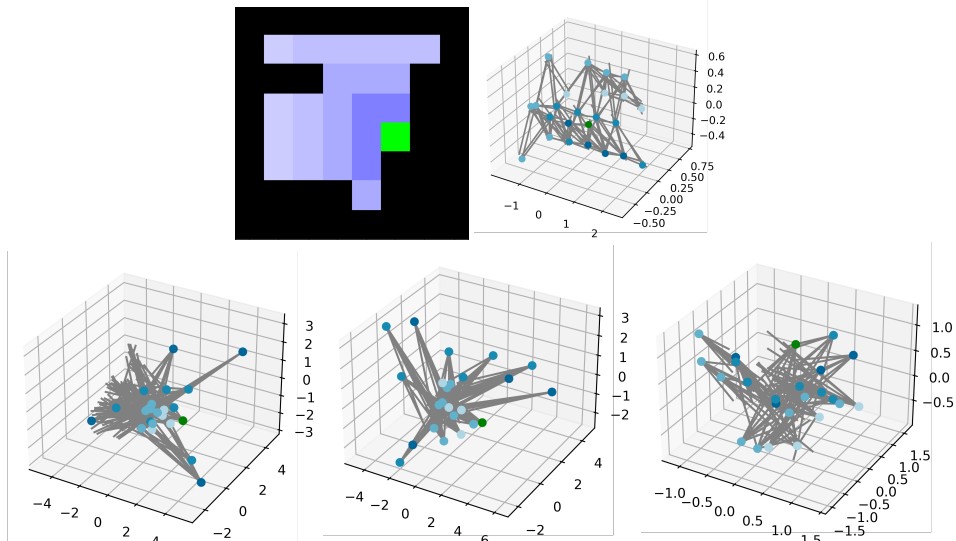

Figure 3: **Top:** A test maze (*left*) and the PCA projection of its TransE state embeddings (*right*), colour-coded by distance to goal (in green). **Bottom:** PCA projection of the XLVIN state embeddings after passing the first (*left*), second (*middle*), and ninth (*right*) level of the continual maze.

**Freeway, Alien, Enduro and H.E.R.O.** Lastly, the average clipped reward of the Atari agents across the first million transitions can be visualised in Figure 2. From the inception of the training, the XLVIN model explores and exploits better, consistently remaining ahead of the baseline PPO model in the low-data regime (matching it in the latter stages of Enduro). In H.E.R.O., XLVIN is also the first agent to break away from the performance plateau, towards the end of the 1,000,000 transitions. The fact that the executor was transferred from randomly generated graphs (Appendix E) is a further statement to XLVIN's robustness.

On all four games, ATreeC consistently trailed behind XLVIN during the first half of the training, and on Enduro, it underperformed even compared to the PPO baseline, indicating that overreliance on scalar predictions may damage low-data performance. It empirically validates our observation of the potential negative effects of algorithmic bottlenecks at low-data regimes.

### 4.3 Qualitative results

The success of XLVIN hinges on the appropriate operation of its two modules; the transition model $T$ and the executor GNN $X$. In this section, we qualitatively study these two components, hoping to elucidate the mechanism in which XLVIN organises and executes its plan.

To faithfully ground the predictions of $T$ and $X$ on an underlying MDP, we analyse a pre-trained XLVIN agent with a CNN encoder and $K = 4$ executor steps on randomly-generated $8 \times 8$ grid-world environments. We chose grid-worlds because $V^\star$ can be computed exactly. We train on mazes of progressively increasing difficulty (expressed in terms of their *level*: the shortest-path length from the start state to the goal). We move on to the next level once the agent solves at least $95\%$ of the mazes from the current level. On these environments, we generally found that XLVIN is competitive with implicit planners that are aware of the grid-world structure. See Appendix F for details, including the hyperparameters of the XLVIN architecture.

**Projecting the embeddings** We begin by qualitatively studying the embeddings learnt by the encoder and transition model. At the top row of Figure 3, we (*left*) colour-coded a specific held-out $8 \times 8$ maze by proximity to the goal state, and (*right*) visualised the 3D PCA projections of the "pure-TransE" embeddings of these states (prior to any PPO training), with the edges induced by the transition model. Such a model merely seeks to organise the data, rather than optimise for returns: hence, a grid-like structure emerges.

At the bottom row, we visualise how these embeddings and transitions evolve as the agent keeps solving levels; at levels one (*left*) and two (*middle*), the embedder learnt to distinguish all 1-step and 2-step neighbours of the goal, respectively, by putting them on opposite parts of the projection space. This process does not keep going, because the agent would quickly lose capacity. Instead, by the time it passes level nine (*right*), grid structure emerges again, but now the states become partitioned by proximity: nearer neighbours of the goal are closer to the goal embedding. In a way, the XLVIN agent is learning to reorganise the grid; this time in a way that respects shortest-path proximity.

$V^\star$ **predictibility** We hypothesise that the encoder function $z$ is tasked with learning to map raw states to a latent space where the executor GNN can operate properly, and then the GNN performs VI-aligning computations in this latent space. We provide a qualitative study to validate this: for all positions in held-out $8 \times 8$ test mazes, we computed the ground-truth values $V^\star(s)$ (using VI), and performed linear regression to test how accurately they are decodable from the XLVIN state embeddings before $(\vec{h}_s)$ and after $(\vec{\chi}_s)$ applying the GNN. We computed the $R^2$ goodness-of-fit measure, after passing each of the ten training levels. Our results (Figure 4) are strongly in support of the hypothesis: while the embedding function $(z(s) = \vec{h}_s$; in green) is already reasonably predictive of $V^\star(s)$ ($R^2 \approx 0.85$ on average), after the GNN computations are performed, the recovered state embeddings $(\vec{\chi}_s$; in red) are consistently almost-perfectly linearly decodable to VI outputs $V^\star(s)$ ($R^2 \approx 1$). Hence, the encoder maps $s$ to a latent space from which the executor can perform VI.

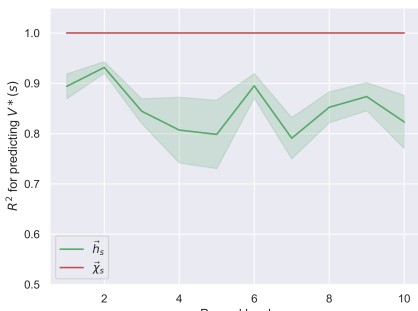

Figure 4: Coefficient of determination from linearly regressing on the state embeddings obtained from the encoder (green) and from the executor (red).

**Algorithmic bottleneck** In formulating the algorithmic bottleneck, we assume that inaccuracies in the scalar values used for VI will have a larger impact on degrading the performance than perturbations in high-dimensional state embeddings inputted to the executor. To faithfully evaluate policy accuracy, we study this sensitivity on the randomly generated MDPs used to train the executor. Here, ground-truth values $V^*$ can be explicitly computed, and we can compare the recovered policy directly to the greedy policy over these values. Hence, we study the effect of introducing Gaussian noise in the scalar inputs fed to VI compared to introducing Gaussian noise in the high-dimensional latents fed into the XLVIN executor. In Figure 5, we plot the policy accuracy as a function of noise standard deviation, showing that, while XLVIN is unable to predict policies perfectly at zero noise, it quickly dominates the VI's policy predictions once the noise magnitude increases. Besides indicating that imperfections in TransE outputs

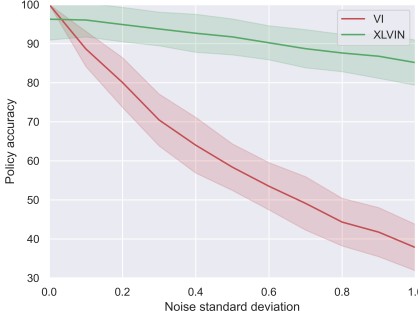

Figure 5: Policy accuracy from introducing Gaussian noise in the scalar input fed into VI (red) and in the embeddings fed into the XLVIN executor (green).

can be handled with reasonable grace, this experiment provides direct evidence of the algorithmic bottleneck: errors in scalar inputs to an algorithm can impact its predictions substantially, while a high-dimensional latent space executor is able to more gracefully handle such perturbations.

## 5 Conclusions

We presented eXecuted Latent Value Iteration Networks (XLVINs), combining recent progress in self-supervised contrastive learning, graph representation learning and neural algorithm execution for implicit planning on irregular, continuous or unknown MDPs. Our results showed that XLVINs match or outperform appropriate baselines, often at low-data or out-of-distribution regimes. The learnt executors are robust and transferable across environments, despite being trained on purely random graphs. XLVINs represent, to the best of our knowledge, one of the first times neural algorithmic executors are used for implicit planning, and they successfully break the algorithmic bottleneck.

## Acknowledgments and Disclosure of Funding

We would like to thank the developers of PyTorch [31]. We specially thank Charles Blundell and Jess Hamrick for the extremely useful discussions. Additionally, we would like to thank all of the anonymous reviewers of our work while it was under submission to ICLR'21, ICML'21 and NeurIPS'21—at every iteration, we received insightful comments that strengthened the paper substantially, and also broadened our own perspective about XLVIN's importance along the way.

AD and JT declare support from the Natural Sciences and Engineering Research Council (NSERC) Discovery Grant, the Canada CIFAR AI Chair Program, collaboration grants between Microsoft Research and Mila, Samsung Electronics Co., Ldt., Amazon Faculty Research Award, Tencent AI Lab Rhino-Bird Gift Fund and a NRC Collaborative R&D Project (AI4D-CORE-06). This project was also partially funded by IVADO Fundamental Research Project grant PRF-2019-3583139727.

PV is a Research Scientist at DeepMind. AD was a Research Scientist Intern at DeepMind while completing this work.

Andreea Deac and Petar Veličković wish to dedicate this paper to their family—for always being by their side, through thick and thin.

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
