

Figure 6: XLVIN model summary with compact dataflow. The individual modules are explained (and colour-coded) in Section 3.1, and the dataflow is outlined in Algorithm 1.

## A    Alternate rendition of XLVIN dataflow

See Figure 6 for an alternate visualisation of the dataflow of XLVIN—which is more compact, but does not explicitly sequentialise the operations of the transition model with the operations of the executor.

## B    Additional description of the Execute function

In Algorithm 2, we provide a symbolic overview of running the executor network $X$ over the local state embedding graph constructed in Algorithm 1.

---

**Algorithm 2:** Forward propagation of the executor

---

**Input**    :State embedding $\vec{h}_s$, executor depth $K$, graph with nodes $\mathbb{S} = \bigcup_{k=0}^{K} \mathbb{S}^k$ and edges $\mathbb{E}$

**Output** :Updated state embedding $\vec{\chi}_s$

**for** $\vec{h} \in \mathbb{S}^k$ **do**
   $\mathcal{N}(\vec{h}) = \{\vec{h}' \mid \exists \alpha.(\vec{h}, \vec{h}', \alpha) \in \mathbb{E}\}$ ;        // Construct neighbourhood of node embedding $\vec{h}$
**end**

$\mathbb{X}^0 = \bigcup_{k=0}^{K} \mathbb{S}^k$;      // We will use $\mathbb{X}^k$ to store the executor embeddings after $k$ steps; initially, $\mathbb{X}^0 = \mathbb{S}$

**for** $k \in [0, K)$ **do**
   **for** $\vec{h} \in \mathbb{X}^k$ **do**
      $\vec{\chi} = X(\vec{h}, \mathcal{N}(\vec{h}))$;        // Run executor on the neighbourhood of node embedding $\vec{h} \in \mathbb{X}^k$
      $M_k(\vec{h}) = \vec{\chi}$;      // Maintain a mapping, $M_k$, from input to output embeddings of $X$ at step $k$
   **end**
   $\mathbb{X}^{k+1} = \{\vec{\chi} \mid \exists \vec{h}.\vec{h} \in \mathbb{X}^k \wedge M_k(\vec{h}) = \vec{\chi}\}$ ;      // $\mathbb{X}^{k+1}$ consists of all outputs of $M_k$
   **for** $\vec{\chi} \in \mathbb{X}^{k+1}$;      // Rebuild neighbourhoods for node embeddings in $\mathbb{X}^{k+1}$
   **do**
      $\mathcal{N}(\vec{\chi}) = \{\vec{\chi}' \mid \exists \vec{h} \exists \vec{h}'.\vec{h} \in \mathbb{X}^k \wedge \vec{h}' \in \mathbb{X}^k \wedge M_k(\vec{h}) = \vec{\chi} \wedge M_k(\vec{h}') = \vec{\chi}' \wedge \vec{h}' \in \mathcal{N}(\vec{h})\}$
   **end**
**end**
$\vec{\chi}_s = M_{K-1}(\dots M_1(M_0(\vec{h}_s))\dots)$ ;      // To recover $\vec{\chi}_s$, follow the mappings $M_k$ starting from $\vec{h}_s$

---

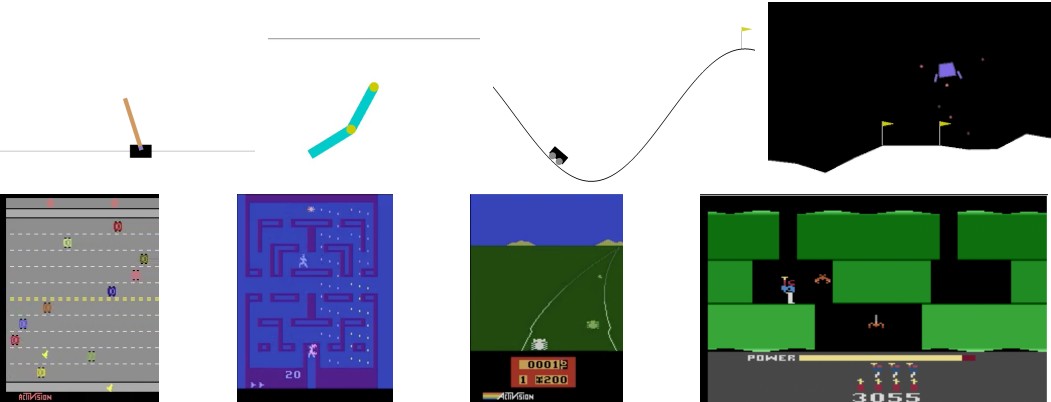

Figure 7: The eight environments considered within our evaluation: continuous control environments (CartPole-v0, Acrobot-v1, MountainCar-v0, LunarLander-v2) and pixel-based environments (Atari Freeway, Alien, Enduro and H.E.R.O.).

## C   Environments under study

We provide a visual overview of all eight environments considered in Figure 7.

**CartPole**   The CartPole environment is a classic example of continuous control, first proposed by [5]. The goal is to keep the pole connected by an un-actuated joint to a cart in an upright position. Observations are four-dimensional vectors indicating the cart's position and velocity as well as pole's angle from vertical and pole's velocity at the tip. Actions correspond to staying still, or pushing the engine forwards or backwards. The agent receives a fixed reward of $+1$ for every timestep that the pole remains upright. The episode ends when the pole is more than 15 degrees from the vertical, the cart moves more than $2.4$ units from the center or by timing out (at 200 steps), at which point the environment is considered solved.

**Acrobot**   The Acrobot system includes two joints and two links, where the joint between the two links is actuated. Initially, the links are hanging downwards, and the goal is to swing the end of the lower link up to a given height. The environment was first proposed by [39]. The observations—specifying in full the Acrobot's configuration—constitute a six-dimensional vector, and the agent is able to swing the Acrobot using three distinct actions. The agent receives a fixed negative reward of $-1$ until either timing out (at 500 steps) or swinging the acrobot up, when the episode terminates.

**MountainCar**   The MountainCar environment is an example of a challenging, sparse-reward, continuous-space environment first proposed by [28]. The objective is to make a car reach the top of the mountain, but its engine is too weak to go all the way uphill, so the agent must use gravity to their advantage by first moving in the opposite direction and gathering momentum. Observations are two-dimensional vectors indicating the car's position and velocity. Actions correspond to staying still, or pushing the engine forward or backward. The agent receives a fixed negative reward of $-1$ until either timing out (at 200 steps) or reaching the top, when the episode terminates.

**LunarLander**   The LunarLander task concerns rocket trajectory optimization—a classic topic in optimal control. It concerns navigating a spaceship in two dimensions to a landing pad (at coordinates $(0, 0)$). Successful landing can be achieved by firing the ship's engines, however this expenses fuel and therefore must be done in a parsimonious manner. The observations are eight-dimensional vectors that include the spaceship's coordinates, velocity, angle of attack, angular velocity, and whether either of its two legs are in ground contact. Actions correspond to firing the main engine, firing one of the two side engines, or idling. The agent receives shaped negative reward corresponding to its distance to the landing pad, and the magnitude of its velocity and angle. Further, fixed negative rewards are incurred whenever the engines are fired (more so for the main engine than the side engines). The agent receives shaped positive rewards of $+10$ whenever its legs make contact with the ground, and a

Table 2: Mean scores for CartPole-v0 after training, averaged over 100 episodes and five seeds. Baseline CartPole results reprinted from [42].

| CartPole-v0 | 100 trajectories | Only 10 trajectories |
|---|---|---|
| REINFORCE | 23.84 $\pm$ 0.88 | - |
| WM-AE | 114.47 $\pm$ 17.32 | - |
| LD-AE | 154.73 $\pm$ 50.49 | - |
| DMDP-H ($J = 0$) | 72.81 $\pm$ 20.16 | - |
| PRAE, $J = 5$ | **171.53** $\pm$ 34.18 | - |
| PPO | - | 104.6 $\pm$ 48.5 |
| XLVIN-R | - | **199.2** $\pm$ 1.6 |
| XLVIN-CP | - | **195.2** $\pm$ 5.0 |

reward of either $+100$ or $-100$ upon completing the episode, dependent on whether landing on the landing pad was successful.

**Freeway**    Freeway is a game for the Atari 2600, published by Activision in 1981, where the goal is to help the chicken cross the road (by only moving vertically upwards or downwards) while avoiding cars. It is a standard part of the Atari Learning Environment and the OpenAI Gym. Observations in this environment are the full framebuffer of the Atari console while playing the game, which has been appropriately preprocessed as in [27]. Actions correspond to staying still, moving upwards or downwards. Upon colliding with a car, the chicken will be set back a few lanes, and upon crossing a road, it will be teleported back at the other side to cross the road again (which is also the only time when it receives a positive reward of $+1$). The game automatically times out after a fixed number of transitions.

**Enduro**    Enduro is a game for the Atari 2600, published by Activison in 1983. The goal of the game is to complete an endurance race, overtaking a certain number of cars each day of the race to continue to the next day. It is a standard part of the Atari Learning Environment and the OpenAI Gym. Observations in this environment are the full framebuffer of the Atari console while playing the game, which has been appropriately preprocessed as in [27]. This game is one of the first games with day/night cycles as well as weather changes which makes it particularly visually rich. There are nine different actions we can take in this environment corresponding to staying still as well as accelerating, decelerating, moving left/right and combinations of two of them.

**Alien**    Alien is a game for the Atari 2600, published by 20th Century Fox in 1982. The goal of the game is to destroy the alien eggs laid in the hallways (similar to the pellets in Pac-Man) while running away from three aliens on the ship. It is a standard part of Atari Learning Environment and the OpenAI Gym. Observations in this environment are the full framebuffer of the Atari console while playing the game, which has been appropriately preprocessed as in [27]. There are 18 different actions we can take in this environment corresponding to staying still, firing the flamethrower and moving or firing the flamethrower in eight directions.

**H.E.R.O.**    H.E.R.O. is a game from Atari 2600, whose goal is to navigate through a mine, clearing obstacles and destroying enemies on the way, in order to rescue a miner at the end of each level. Similarly to Alien, observations in this environment are the full framebuffer of the Atari console while playing the game, which has been appropriately preprocessed as in [27] and the action space is formed of 18 different actions.

## D    Additional CartPole results

In Table 2, we provide a comparison between XLVIN and several baselines from [42].

## E    Synthetic graphs

Figure 8 presents the two kinds of synthetic graphs used for pretraining the GNN executor.

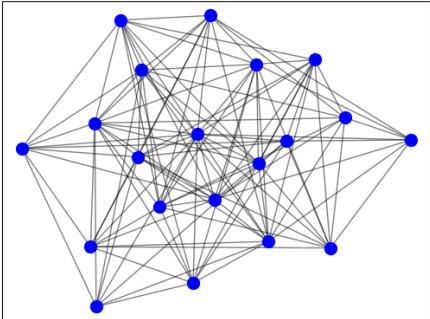 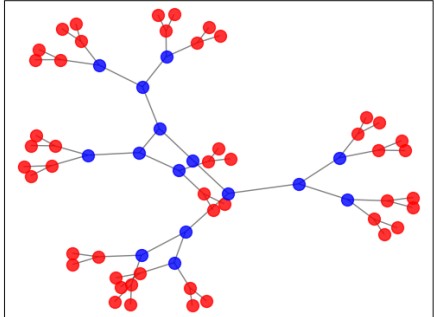

Figure 8: Synthetic graphs constructed for pre-training the GNN executor: random deterministic (20 states, 8 actions) (**left**) and CartPole (**right**)

In most cases, we pre-train the executor using randomly generated deterministic graphs (left): for $|\mathcal{S}| = 20$ states and $|\mathcal{A}| = 8$ actions, we create a $|\mathcal{S}|$-node graph. For each state-action pair we select, uniformly at random, the state it transitions to, deterministically. We sample the reward model using the standard normal distribution. Overall, the graphs are sampled as follows:

$$\tilde{T}(s, a) \quad \sim \quad \text{Uniform}(|\mathcal{S}|) \tag{8}$$

$$P(s' \mid s, a) \quad = \quad \begin{cases} 1 & s' = \tilde{T}(s, a) \\ 0 & \text{otherwise} \end{cases} \tag{9}$$

$$R(s, a) \quad \sim \quad \mathcal{N}(0, 1) \tag{10}$$

These $k$-NN style graphs do not assume upfront any structural properties of the underlying MDP, and are a good prior distribution for evaluating the performance of XLVIN.

For CartPole-style environments, we attempt a different type of graph (right). It is a binary tree, where red nodes represent nodes with reward 0, and blue nodes have reward 1. This aligns with the idea that going further from the root, which is equivalent with taking repeated left (or right) steps, leads to being more likely to fail the episode.

We also attempt using the CartPole graph for pre-training the executor for the other two continuous-observation environments (MountainCar, Acrobot). Primarily, the similar action space of the environments is a possible supporting argument of the observed transferability. Moreover, MountainCar and Acrobot can be related to a inverted reward graph of CartPole, with more aggressive combinations left/right steps often bringing a higher chance of success.

## F  Maze results

As described in the main text, in order to qualitatively assess the transition and executor modules in XLVIN, we evaluated them on a known, fixed and discrete MDP—where optimal values $V^\star(s)$ can be trivially computed. Accordingly, we use the $8 \times 8$ and $16 \times 16$ grid-world mazes proposed by [40]. The observation for this environment consists of the maze image, the starting position of the agent and the goal position. Every maze is associated with a *difficulty* level, equal to the shortest path length between the start and the goal.

Using this concept, we formulate the *continual maze* task: the agent is, initially, trained to solve only mazes of difficulty level 1. Once the agent reaches 95% success rate on the last 1,000 sampled episodes of level $d$, it advances to level $d + 1$ (without observing level $d$ again). If the agent fails to reach 95% within 1,000,000 trajectories, it is not allowed to progress. After each passed difficulty, the agent is evaluated by computing its success rate on held-out test mazes.

Given the grid-world structure, our encoder for the maze environment is a three-layer CNN computing 128 latent features and 10 outputs. The transition function is a three-layer MLP with layer

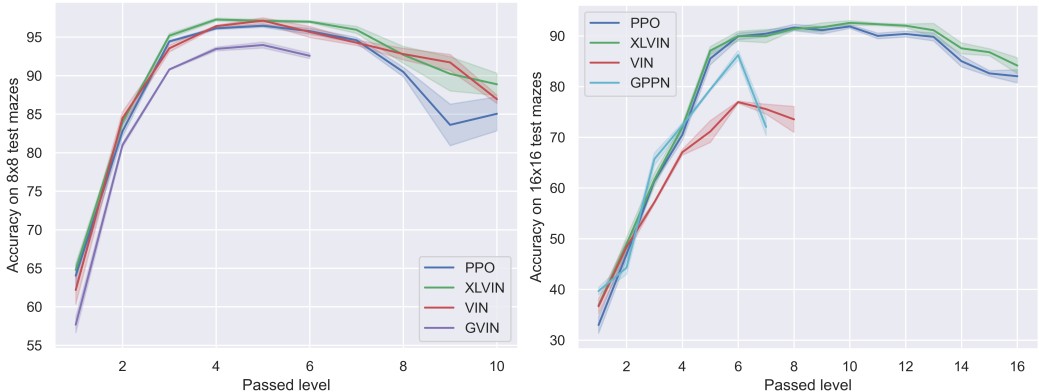

Figure 9: Success rate on $8 \times 8$ (**left**) and on $16 \times 16$ (**right**) held-out mazes obtained after passing each level of their respective train mazes. Cut-off curves imply **failure** to pass a difficulty level.

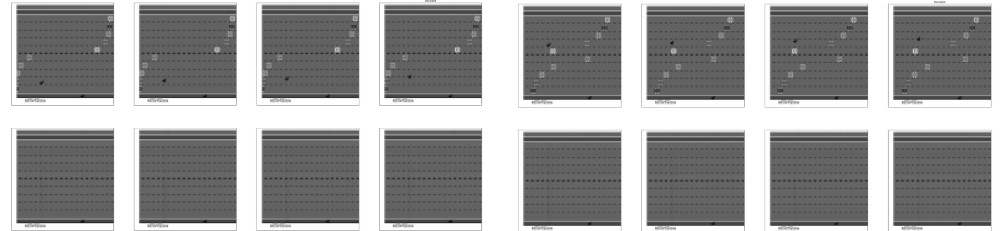

Figure 10: Freeway frames (**above**) and reconstructions (**below**) using a VAE-style world model.

normalisation [4] after the second layer, computing 128 hidden features. We apply the executor until depth $K = 4$, with layer normalisation applied after every step.

Beyond its use for qualitative evaluation, we also perform a comparison of XLVINs against several standard implicit planners in this space (including (G)VIN and GPPN). The results are summarised in Figure 9, and indicate that XLVIN is competitive with all other models, while not making any upfront assumptions about the dynamics of the environment.

## G   Pixel-based world models

XLVIN is, in principle, agnostic to the choice of transition model. We chose a latent-space transition model in the style of TransE because this aligned the closest with the ATreeC baseline, which also used a latent-space transition model. World models that predict full observations are also plausible.

We attempt replacing our Atari transition model with a variant that learns representations through pixel-based reconstructions (using a VAE objective, as done by [19]). We found that representations obtained in this way were not useful; we observed that most of our state encodings converged to a fixed-point, and that the pixel-space reconstructions completely ignored the foreground observations (see Figure 10). This aligns with prior investigations of VAE-style losses on Atari, which found they tend to overly focus on reconstructing the background and were less predictive of RAM state than latent-space models, as well as randomly-initialised CNNs [3]. This comparison stands in favour of our approach to using a transition model optimised purely in the lower-dimensional latent space.

## H   Compute details

We used one V100 GPU from an internally provided cluster for training our model on the Atari environments, for which the training time for one seed per environment was always less than 24 hours. For the classical control and navigation tasks, a 2.7GHz i7 CPU was used.

We used the OpenAI Gym [9] for access to environments, PytorchRL [25] for the PPO implementation and encoder parameters and OpenAI Baselines [12] for environment wrapper capabilities. All of the above are licensed under the MIT license.

## I  Potential societal impact

Our work studies fundamental insights related to implicit planners. The problem of improving data efficiency, while building better plans is highly important for real world applications. However, this work does not explicitly focus on the engineering efforts for such applications or implications. Instead, we analyse the problem from a theoretical and empirical angle, first identifying bottleneck issues in prior art and then empirically verifying the effects of alleviating the bottleneck on classical control and standard game-playing benchmarks. Therefore, we consider direct societal impact not to be applicable in this setting.