# OpenReview forum: "Neural Algorithmic Reasoners are Implicit Planners"
_NeurIPS.cc/2021/Conference — NeurIPS 2021 Spotlight_

### Official Review · Reviewer_UvSf · 2021-06-28

**Rating:** 7
**Confidence:** 3

**Summary:**

In the context of implicit planning algorithms, the paper proposes to replace exact value iteration (VI) computations with a learned graph neural network that is trained to perform VI computations in an abstract latent space.

The paper shows competitive performance on several benchmark datasets using this approach, against a PPO baseline, and an implicit planning algorithm using exact VI.


**Limitations And Societal Impact:**

Yes

**Main Review:**

The paper argues that performing exact VI computations in an implicit planning setup suffers from an "algorithmic bottleneck". They argue that this algorithmic bottleneck occurs when the exact VI computations are performed on poor initial estimates of the value function.

To remedy this "algorithmic bottleneck" the paper proposes to instead perform (approximate) VI computations in a latent space using a pre-trained Graph Neural Network (GNN). The GNN consists of 3 parts, an encoder, an update function, and a decoder, and is pre-trained to perform approximate VI on a dataset of randomly sampled Markov Decision Processes, after which only the update function is kept and frozen.

Given this GNN, the "executor", that can perform approximate VI in a latent space, the paper uses a fairly standard implicit planning setup. A learned encoder maps the environment state into a latent space, and a learned transition function maps latent states and actions to the next (expected) latent state. Given this tree of latent states, the executor performs its approximate VI computations and finally, a learned policy and value network outputs the policy and value function.

The paper is exceptionally well written. Especially section 2 gives the reader an excellent overview of the relevant material needed to understand the contributions of the paper. The paper shows compelling results demonstrating that it's better than a solid baseline (PPO) and more importantly, ATreeC, which is the method they directly built on by replacing the exact VI computations.

I have a few comments that I think could make the paper even stronger.

1) The paper argues that the algorithmic bottleneck is a problem, but doesn't really offer any evidence of this, except in the full experiments where they show their proposed method is better than the one using the exact VI computations. Further, the paper doesn't give any intuition as to why performing the VI computations in a high dimensional latent space might avoid the algorithmic bottleneck. I would greatly appreciate a small-scale experiment that directly shows the effect of the "algorithmic bottleneck" and how doing (approximate) VI in a higher dim latent space avoids it. The authors could use the simple MDPs they use to train the executor, and inject noise in the value functions and show how their learned executor is more robust to noise than the direct VI equations. This along with a few sentences on the intuition why doing approximate VI in a higher dim space is more robust would greatly improve the paper in my opinion.

2) It was not entirely clear to me how the whole system was trained. In particular, I find it hard to understand how the encoding function and transition functions are trained. The loss in eq. 4 seems to indicate that they're trained entirely offline on (S,A,S') triplets, but that can't be true since then the learned embeddings wouldn't align with the one used by the executor. It would be very helpful if the authors included the joint loss that is optimized with PPO, and/or an algorithm describing the training procedure (training the executor first, then freezing, etc.).

A few minor things:
 - I'd have liked a small discussion of how well (or not) the approach handles multi-modal MDPs. Computing an expected state over a multi-modal outcome seems like a poor approach.
 - Line 156: "... and do not project our state to a low-dimensional space": well, you do, with your encoder function. I know what you mean, but this sentence is kind of confusing.

**Time Spent Reviewing:**

6

---

> ### Author Response · Authors · 2021-08-11
> **Reply to Reviewer UvSf**
>
> Thank you for the comprehensive review and the kind words about our writeup and contributions, and also for the insightful suggestions!
>
> We address your comments in turn, with faith that they have made our contributions stronger.
>
> Please also see our other responses, which detail additional experiments that we have performed during the rebuttal phase. We welcome any and all feedback!
>
> ### **Qualitative experiments on the algorithmic bottleneck**
> We indeed haven’t, at any point, explicitly demonstrated the algorithmic bottleneck in our original manuscript. We wholeheartedly agree that such a demonstration would be highly valuable, and have set up a follow-up experiment exactly guided by your suggestion. Thank you!
>
> For the randomly generated MDPs used to train our executor, we have compared the policy accuracy (w.r.t. greedy policy over the optimal value function $V^\star$), when
>
> * introducing Gaussian noise in the scalar inputs fed to VI,
> * introducing Gaussian noise in the high-dimensional latents fed into the XLVIN executor.
>
> We plot the policy accuracy as a function of noise standard deviation, showing that, while XLVIN is unable to predict policies perfectly at zero noise, it quickly dominates the VI’s policy predictions once the noise magnitude increases.
>
> Our results can be found here:
>
> https://anonymous.4open.science/r/XLVIN-figs-NeurIPS-870B/acc_sigma.png
>
> This is direct evidence of the algorithmic bottleneck: errors in scalar inputs to an algorithm can impact its predictions substantially, while a high-dimensional latent space executor is able to more gracefully handle such perturbations.
>
> We will include this experiment, along with a discussion, in the revised paper!
>
> ### **Clarifying the training regime**
> Thank you for bringing this point to our attention! You are correct to identify that the TransE component is trained on-line, and we will clarify the policy training setting carefully in a revised version of the paper.
>
> As a brief overview:
> - At each step our model samples several on-policy trajectories (with multiple parallel actors acting for a fixed number of steps)
> - Based on the transitions in these rollouts, we evaluate the PPO and TransE losses. Negative sample states for TransE are randomly sampled from the rollouts.
> - The total loss function to optimise is then $L_\text{PPO} + \lambda L_\text{TransE}$ where $\lambda = 0.001$ for our experiments.
>
> We will also aim to include the full training algorithm, which would also include the pre-training and freezing of the executor function.
>
> ### **On MDPs with multimodal outcomes**
> Thank you for raising this point! We believe that multi-modal outcome MDPs are highly relevant, and there are certain limitations when handling them with the expected embedding over all possible outcomes. Specifically, while taking expectations is an operation that aligns well with VI computations, such “averaged” latent states may not be trivially further expandable, should we wish to plan further from them.
>
> One possible remedy we propose for this is employing a probabilistic (e.g. variational) transition model from which we could repeatedly sample concrete latents, and we will include this discussion in a revised paper.
>
> ### **On projecting to a low-dimensional space**
> Very well spotted! We will carefully rephrase this to make the desired meaning more explicit.

---

> > ### Comment · Reviewer_UvSf · 2021-08-23
> > **Response**
> >
> > Thank you for your response.
> >
> > Thank you for the additional experiments re. the algorithmic bottleneck. I think it's a very strong result that really help with the papers main argument. I also think it's quite surprising! Why is a learned approximation more robust to noise than the algorithm it's trained to mimic? Any intuition you have on this would be greatly appreciated. Is it because noise is implicitly added because you're using SGD? Another (less exciting) possible hypothesis is that it's because the noise you add is IID, so adding multiple IID samples to the latent space their effects kind of cancel out. Maybe you could try sampling a single noise sample and adding that to all the values of the latent space vector, and see how that compares.
> >
> > I'll retain my inital review, and hope to see this get published.

---

> > > ### Author Response · Authors · 2021-08-30
> > > **Follow-up experiment for Reviewer UvSf**
> > >
> > > Thank you for the acknowledgement and the additional useful points raised!
> > >
> > > We have now performed the follow-up experiment you proposed (with non-i.i.d. noise), and we observed very minor (on the order of 0.5%) reductions in policy accuracy compared to the i.i.d noise case. Hence our method is robust even in the face of lower-entropy perturbations.
> > >
> > > Regarding your point about robustness, we generally agree with your intuition. The fact that our executor network is jointly trained with an encoder network (which carries inputs to VI into the latent space), leveraging stochastic gradient descent (which implicitly injects noise into the system), implies that the processor has a good chance to be robust to perturbations in the encoder -- and hence, perturbations in its latent space.
> > >
> > > We are happy to incorporate this discussion into the paper!

---

### Official Review · Reviewer_8TgE · 2021-07-15

**Rating:** 8
**Confidence:** 5

**Summary:**

This paper proposes a new implicit planner module, XLVINs, for end-to-end policy learning, which mimics the value iteration planning procedure with a pre-trained graph convolution network and overcomes the existing limitations of previous VI-style planning modules. This paper carefully investigates the existing literature with a thorough analysis of the algorithmic limitations of previous VI modules. The experiments also provide a complete understanding of the proposed method.

**Limitations And Societal Impact:**

There are some minor comments.

Regarding the TransE, it seems a bit unjustified since it seems that even if we simply train a neural network to predict the next state (e.g., a transition model, or a world model), things would still work without any changes? Is there any specific reason for using TransE other than its simplicity?

Finally, I definitely feel the experiment scenarios are sufficient in the current paper. If possible, the paper can be even stronger if results on more Atari games can be presented.

**Main Review:**

The paper is extremely well written with lots of insightful discussions and explanations. It is really a great pleasure to read this paper and everything is so easy to follow under its current form. The discussion of algorithmic bottleneck is insightful.

From a technical perspective, the paper proposes to use a TransE style latent model and adopt a GCN-based VI module. These two techniques directly address the existing limitations of previous works, which look clearly novel to me. I like the idea of pre-training a VI-module using random graphs and was surprised to see it is really working ---- particularly the experiment where the ground truth information is recovered.

In general, I like this paper and feel happy to see it gets accepted.

**Time Spent Reviewing:**

1

---

> ### Author Response · Authors · 2021-08-11
> **Reply to Reviewer 8TgE**
>
> Thank you very much for the very kind review, and your constructive comments!
>
> We address your comments in turn, with faith that they have made our contributions stronger.
>
> Please also see our other responses, which detail additional qualitative experiments that we have performed during the rebuttal phase. We welcome any and all feedback!
>
> ### **Justification of TransE**
> It is true that our framework is in principle agnostic to the choice of transition model. We chose a latent-space transition model in the style of TransE because this aligned the closest with the ATreeC baseline (which also used a latent-space transition model).
>
> World models that predict full observations are also plausible, but align somewhat weaker with our latent-space executor, and in some cases (e.g. Atari Freeway) we observed that naïvely using a VAE-style world model can lead to unfavourable reconstructions (e.g. models that only learn to predict a static background, without really reconstructing any of the foreground).
>
> This is in line with previous findings on Atari state representation learning (e.g. [1]) and ultimately contributed to our decision to use a latent-space transition model.
>
> We will aim to include this discussion, along with sample state reconstructions on Freeway from a VAE model, in our supplementary material.
>
> ### **Additional results on Atari H.E.R.O.**
> As per your suggestion, we provide further comparisons of XLVIN against ATreeC and PPO on low-data H.E.R.O., which is a sparse-reward, visually complex Atari game.
>
> Our results in terms of clipped rewards are summarised in the following figure:
>
> https://anonymous.4open.science/r/XLVIN-figs-NeurIPS-870B/Hero_scores.png
>
> The results are in line with our algorithmic bottleneck hypothesis: XLVIN makes substantial progress in the early phases of training (first ~200k iterations), quickly reaching the first plateau. ATreeC, in comparison, is slowest of the three models to reach this point, and remains unstable afterwards. XLVIN is also consistently the first model to break away from this plateau (towards the end of the 1M transitions).
>
> We will include these experiments and discussions in a revised paper!
>
> [1] Anand _et al._, Unsupervised State Representation Learning in Atari; NeurIPS’19

---

> > ### Comment · Reviewer_8TgE · 2021-08-17
> > **Response acknowledged**
> >
> > Thanks for the additional results, which I would be definitely happy to see in the final revision.

---

### Official Review · Reviewer_wygb · 2021-07-16

**Rating:** 7
**Confidence:** 3

**Summary:**

This work extends the value iteration network family of reinforcement learning agents to a much less constrained form to avoid the "algorithmic bottleneck". The proposed model XLVIN creates a plannable representation of the state space by 1) first learning an embedding space for the state transitions via a TransE (or contrastive) approach and 2) then aggregating several layers of look-ahead state embeddings via a pretrained GNN. The policy and value functions are parameterized on top of the plannable representation. Empirically XLVINs provide a significant boost on the stochastic control tasks and Atari games in the low-data regime.

**Limitations And Societal Impact:**

Yes.

**Main Review:**

Pros:
- This paper provides a solution to the issues presented in previous work along this line:
  - Not all high-dimensional state spaces can be mapped properly to a lower-dimensional space for VIN-like updates.
  - The implicit planning in previous work is prone to error, especially from the incorrect estimation of the reward and state values.
- The improvements compared with existing baselines in low-data regimes are significant.
- The paper is well written, and the experiments are solid with several extra studies beyond standard ones.

Cons:
- Since the complexity of the proposed method scales with O(|A|^K), the model can only look ahead for a very small number of steps.
  - Why would a small K help build a better state representation? For atari games, it seems that looking 1-2 steps ahead into the future won't provide a decent gain.
- The discussion of how the quality of the learned TransE state embeddings will affect the final performance is not included in the evaluation section. Such sensitivity analysis will be helpful in seeing how XLVINs can generalize to more complicated state spaces (e.g., molecule space) where TransE style embeddings are hard to learn.

**Time Spent Reviewing:**

3

---

> ### Author Response · Authors · 2021-08-11
> **Reply to Reviewer wygb**
>
> Thank you for your careful review, the kind words about the benefits of our method, and all of your suggested improvements.
>
> We address your comments in turn, with faith that they have made our contributions stronger.
>
> Please also see our other responses, which detail additional experiments that we have performed during the rebuttal phase. We welcome any and all feedback!
>
> ### **Computational complexity, and the effect of $K$ on Atari**
> Thank you for raising these points! We fully acknowledge the scalability properties of our method as proposed, and already have a detailed discussion in the paper of ways in which the complexity can be improved (e.g. learning or distilling rollout policies for targeted expansions).
> While it is true that this limits the effective maximal possible expansion of our model, we found that in practice, adding further steps lead to plateaus in performance. This phenomenon was also observed in several related papers that study implicit planning on Atari — in particular, the paper introducing our ATreeC baseline [1].
>
> Even if these are shallow trees of states, we anticipate a compounding effect from optimising TransE together with PPO’s value/policy heads. Optimising TransE leads to trees that more faithfully represent the neighbouring states’ latents, while optimising the values predicted from these latents implies that they will give immediate context on which actions are locally optimal — this can, in turn, facilitate more targeted exploitation during early phases of training, especially with respect to a purely reactive baseline (PPO).
>
> Note that all of the above benefits hold for ATreeC as well; however, the ATreeC is bottlenecked on having to predict correct value/reward models before it can reliably exploit the algorithm outputs, leading to inefficiencies in the low-data regime (which we explicitly address).
>
> We will include this discussion in a revised paper!
>
> ### **Sensitivity to TransE embedding quality + demonstrating the algorithmic bottleneck**
> We fully agree that it is useful to be mindful of complicated state spaces, such as ones pertaining to molecular generation. In light of this, we wanted to demonstrate the effects of injecting noise into the embeddings that are produced by TransE on the policy accuracy.
>
> In order to faithfully evaluate policy accuracy, we decided to study this sensitivity on the randomly generated MDPs used to train our executor. Here, ground-truth values $V^\star$ can be explicitly computed, and we can compare the recovered policy directly to the greedy policy over these values.
>
> Once the transition model is fully expanded, the outputs of TransE are only ever directly fed into the executor, and are not used in other ways. Therefore, it made sense to study the effect of perturbations on the high-dimensional latents fed into the XLVIN executor.
>
> We study the effect of:
>
> - introducing Gaussian noise in the scalar inputs fed to VI,
> - introducing Gaussian noise in the high-dimensional latents fed into the XLVIN executor.
>
> We plot the policy accuracy as a function of noise standard deviation, showing that, while XLVIN is unable to predict policies perfectly at zero noise, it quickly dominates the VI’s policy predictions once the noise magnitude increases.
>
> Our results can be found here:
>
> https://anonymous.4open.science/r/XLVIN-figs-NeurIPS-870B/acc_sigma.png
>
> Besides indicating that imperfections in TransE outputs can be handled with reasonable grace, this experiment provides direct evidence of the algorithmic bottleneck: errors in scalar inputs to an algorithm can impact its predictions substantially, while a high-dimensional latent space executor is able to more gracefully handle such perturbations.
>
> We will include this experiment, along with a relevant discussion, in the revised paper!
>
> [1] Farquhar _et al._, TreeQN and ATreeC: Differentiable Tree-Structured Models for Deep Reinforcement Learning. ICLR’18

---

> > ### Comment · Reviewer_wygb · 2021-08-25
> > **Response acknowledged**
> >
> > Thanks for providing the analysis and adding experiments in such a short time! I'd be happy to see the paper accepted.

---

### Official Review · Reviewer_Li46 · 2021-07-19

**Rating:** 6
**Confidence:** 4

**Summary:**

This submission proposes eXecuted Latent Value Iteration Networks (XLVINs) to relax the assumptions of Value Iteration Networks. The method infers the dynamics of the MDP using known representation learning techniques and uses the inferred dynamics to construct a graph representation which is used for planning. The method allows continuous state spaces and unlike VINs, not restricted to grid-world.

**Limitations And Societal Impact:**

Yes

**Main Review:**


Note:
I reviewed this paper for ICML 2021. The method is identical as far as I can tell. The writing quality has been improved at some places. In terms of experiments, the authors have removed known MDP maze experiments and have added one continuous space environment, LunarLander-v2.


Strengths:
- The paper tackles an important problem of planning under uncertain maps which is of interest to the community.
- The method incorporates several prior techniques to tackle planning from pixel space which makes it complicated, but it is described in sufficient detail at most places.


Weakness:
- The paper aims to improve over VINs by tackling continuous space and pixel space environments. However, while doing so it seems the proposed method isn't as effective as VINs at discrete state planning under known MDPS. Instead of adding larger maze size experiments, the authors chose to remove all maze experiments. If the method isn't effective at known MDP discrete space planning, this should be acknowledged in the paper.
- For continuous state space tasks, the authors only conduct experiments in only 4 simple gym games and for pixel space observations, only 3 Atari games. The motivation of using these benchmarks for testing planning algorithms is unclear. It would make more sense to test the proposed method for planning in high dimensional continuous state space 3D manipulation or navigation tasks. In any case, the experimental evaluation needs to be more comprehensive in my opinion to establish the effectiveness of XLVINs.
- The paper misses some relevant related work, for example Universal planning networks (Srinivas et al. ICML 2018), Differentiable MPC for End-to-End Planning and Control (Amos et al., Neurips 2018). It would be useful to discuss how the proposed method differs from these methods.
- The baselines used are weak in my opinion. The proposed model learns a transition model between latent states. It seems more similar to model-based RL methods than to VINs. I would have liked to see empirical comparisons to Universal planning networks (Srinivas et al. ICML 2018) or PlaNet (Hafner et al. ICML 2019).


Update after author response:
The author response has addressed my major concerns. I am increasing my rating accordingly. I still believe larger mazes and robotics applications are better environments to test planning methods rather than a few Atari games.


**Time Spent Reviewing:**

2

---

> ### Author Response · Authors · 2021-08-11
> **Reply to Reviewer Li46**
>
> Thank you for your careful review, recognising some of our improvement efforts, and your suggestions for improvement!
>
> We address your comments in turn, with faith that they have made our contributions stronger.
>
> Please also see our other responses, which detail additional qualitative experiments that we have performed during the rebuttal phase. We welcome any and all feedback!
>
> ### **Maze experiments and comparisons to VIN**
> Before we address this claim from a theoretical point of view, it is important to note that our maze experiments had not been completely removed — they are the foundation of all of our qualitative studies into XLVIN’s properties (leveraging the fact that in such domains, the ground-truth values $V^\star$ can be explicitly computed, allowing us to explicitly reason about our model’s ability to perform VI-like reasoning).
>
> Further, our quantitative results on both the 8x8 and larger 16x16 continual mazes are provided in the supplementary material (Appendix F), and explicitly referred to in the main paper.
>
> The reason why these experiments have been relegated into the appendices is the specific perspective we are offering on XLVINs in this paper, through the lens of algorithmic bottlenecks.
>
> It is important to note how VIN can be seen as a special case of an XLVIN that has upfront knowledge about the grid domain. Namely, if we place XLVINs in a grid world, and make the transition model a nonparametric function that merely performs the required shift of the agent’s coordinates in the grid, our GNN executor would then boil down to exactly a convolutional operation over the grid. We make this connection explicit in lines 116—118 of our paper (“For the special case of grid worlds, the neighbours of a grid cell correspond to exactly its neighbouring cells, and hence the rules in Equations 2–3 amount to a convolutional neural network over the grid [32]”)
>
> Hence, VINs correspond to an XLVIN that already has the oracle transition model, and therefore a much easier job fitting this particular domain. Our experiments on grids demonstrate that the generic XLVIN model without such oracles is competitive with VIN, especially if a degree of distribution shift is to be expected.
>
> Conversely, we position our paper on value iteration-like planning in generic domains, where VINs are no longer applicable. We contribute by discovering the algorithmic bottleneck effect in this setting and proposing a direct way to ameliorate it. As such, our natural baseline are not VINs, but the ATreeC family of models (which also encompasses proposals like TreeQN [1] and VPN [2]).
>
> We are happy to make all of the above discussion more explicit in the paper!
>
> ### **Motivation for the chosen environments + new experiments on Atari H.E.R.O.**
> Regarding the environments considered in this work, we claim that several of them are explicitly studied in the context of studies on planning within deep reinforcement learning. For example, Acrobot is one of the environments identified in a recent thorough study [3] to be a highly challenging environment that necessitates planning computations for generalisation. CartPole has been extensively studied in recent work bridging geometric deep learning and planning [4]. Similarly, Freeway, Alien and Enduro have all been identified as Atari games that can benefit from having a strong model of the world — see [1, 2, 5] for recent examples.
>
> As per your suggestion, we provide further comparisons of XLVIN against ATreeC and PPO on low-data H.E.R.O., which is a sparse-reward, visually complex Atari game, also likely necessitating long-range reasoning.
>
> Our results in terms of clipped rewards are summarised in the following figure:
>
> https://anonymous.4open.science/r/XLVIN-figs-NeurIPS-870B/Hero_scores.png
>
> The results are in line with our algorithmic bottleneck hypothesis: XLVIN makes substantial progress in the early phases of training (first ~200k iterations), quickly reaching the first plateau. ATreeC, in comparison, is slowest of the three models to reach this point, and remains unstable afterwards. XLVIN is also consistently the first model to break away from this plateau (towards the end of the 1M transitions).
>
> We will include these experiments and discussions in a revised paper!
>
> ### **Additional related work**
> Thank you for remarking about UPN and Differentiable MPC, which are both relevant approaches that we will cite in our related work. Our aim—to learn a generally applicable reasoning rule within the planner (value iteration), without a specified goal in mind, contrasts with UPN’s predominantly-studied approach of training the planner to optimise a supervised imitation learning objective. Further, UPNs are goal-conditioned, whereas our differentiable executors proved applicable across a wide variety of domains where goals are not known upfront (continuous-state control as well as Atari).
>
> Differentiable MPC is related to our work in terms of aligning to an algorithm, but using the algorithm explicitly (as Differentiable MPC does) often has issues of requiring a bespoke backpropagation rule, and the associated low-dimensional bottlenecks. For these reasons, we chose ATreeC as our bottlenecked baseline. ATreeC is a very natural baseline for our work, since: i) XLVINs generalise ATreeC with respect to the knowledge representation - we use state embeddings computed in VI-like manner instead of scalars, ii) XLVINs generalise ATreeC with respect to the planning algorithm - we use learned VI integrated in a neural network instead of explicitly committing to scalar reward estimates, and iii) ATreeC and XLVINs are
> applicable to the same kinds of environments.
>
> We will update the paper accordingly with the above citations written in.
>
> ### **XLVINs are implicit planners**
> Lastly, we would like to reaffirm that XLVINs are implicit planners -- no explicit discrete planning algorithm is invoked. In contrast, approaches such as PlaNet explicitly perform a planning algorithm in their actors once the required models are learnt, and they do not feature any policy or value networks. Hence, we do not consider XLVIN strongly related to such approaches.
>
> While XLVINs perform planning-like computations within their forward pass, they are optimised principally by gradient descent on a model-free loss (PPO). This, indeed, makes them best aligned with the ATreeC family of models, in terms of bottlenecked baselines.
>
> [1] Farquhar _et al._, TreeQN and ATreeC: Differentiable Tree-Structured Models for Deep Reinforcement Learning. ICLR’18
>
> [2] Oh _et al._, Value Prediction Networks. NeurIPS’17
>
> [3] Hamrick _et al._, On the role of planning in model-based deep reinforcement learning, ICLR’21
>
> [4] van der Pol _et al._, Plannable Approximations to MDP Homomorphisms: Equivariance under Actions, AAMAS’20
>
> [5] Kaiser _et al._, Model Based Reinforcement Learning for Atari, ICLR’20

---

### Author Response · Authors · 2021-08-30
**Thank you to all reviewers!**

We would like to thank all the reviewers for their careful consideration of our work, their insightful comments, and their kind remarks on our responses.

As mentioned, we are very happy to revise the paper incorporating all of the discussion points raised!

---

### Decision · Program_Chairs · 2021-09-27

**Decision:**

Accept (Spotlight)

**Comment:**

The paper proposes proposes eXecuted Latent Value Iteration Networks (XLVINs) to relax the assumptions of Value Iteration Networks.  While most neural architectures in deep RL are not well motivated, this paper builds on an important line of work that treats value representations as implicit planners.  This is very nice work that generalizes previous work.  Well done!